# The Impact of microRNAs in Renin–Angiotensin-System-Induced Cardiac Remodelling

**DOI:** 10.3390/ijms22094762

**Published:** 2021-04-30

**Authors:** Michaela Adamcova, Ippei Kawano, Fedor Simko

**Affiliations:** 1Department of Physiology, Faculty of Medicine in Hradec Kralove, Charles University in Prague, Simkova 870, 500 03 Hradec Kralove, Czech Republic; kawanoip0629@gmail.com; 2Institute of Pathophysiology, Faculty of Medicine, Comenius University, 81108 Bratislava, Slovakia; fedor.simko@fmed.uniba.sk; 33rd Department of Internal Medicine, Faculty of Medicine, Comenius University, 83305 Bratislava, Slovakia; 4Biomedical Research Center, Institute of Experimental Endocrinology, Slovak Academy of Sciences, 84505 Bratislava, Slovakia

**Keywords:** miRNA, RAS, cardiac remodelling, cardiac fibrosis, cardiac hypertrophy

## Abstract

Current knowledge on the renin–angiotensin system (RAS) indicates its central role in the pathogenesis of cardiovascular remodelling via both hemodynamic alterations and direct growth and the proliferation effects of angiotensin II or aldosterone resulting in the hypertrophy of cardiomyocytes, the proliferation of fibroblasts, and inflammatory immune cell activation. The noncoding regulatory microRNAs has recently emerged as a completely novel approach to the study of the RAS. A growing number of microRNAs serve as mediators and/or regulators of RAS-induced cardiac remodelling by directly targeting RAS enzymes, receptors, signalling molecules, or inhibitors of signalling pathways. Specifically, microRNAs that directly modulate pro-hypertrophic, pro-fibrotic and pro-inflammatory signalling initiated by angiotensin II receptor type 1 (AT1R) stimulation are of particular relevance in mediating the cardiovascular effects of the RAS. The aim of this review is to summarize the current knowledge in the field that is still in the early stage of preclinical investigation with occasionally conflicting reports. Understanding the big picture of microRNAs not only aids in the improved understanding of cardiac response to injury but also leads to better therapeutic strategies utilizing microRNAs as biomarkers, therapeutic agents and pharmacological targets

## 1. The Renin–Angiotensin System

The renin–angiotensin system (RAS) has been traditionally associated with the control of electrolyte balance and blood pressure regulation. However, its chronic activation may impair cardiac function and structure both via hemodynamic overload as well as direct trophic action on cardiac and vascular tissues. The deleterious effect of chronic RAS activation is reflected by the success of RAS inhibition in a variety of cardiovascular pathologies [1,2,3,4,5,6,7].

### 1.1. Classical Pathways

The classical pathway of the RAS is activated by the renin-induced cleavage of liver-synthesized angiotensinogen to the primary effector molecule, angiotensin II (Ang II). Ang II is the principal player in the RAS, whose effect is mainly mediated by binding to angiotensin receptor type I (AT1R); however, angiotensin receptor type II (AT2R) and other G protein-coupled receptors (GPCRs) may be involved [8]. The crucial Ang II effects are mediated by AT1R are circulatory volume and vascular tone regulation, hypertrophic and fibrotic target organ remodelling, reactive oxygen species (ROS) production, and the modulation of immune response and inflammation [9,10,11,12]. The function of AT2R is not fully understood, but it is considered to antagonize the effect of AT1R by promoting vasodilation along with antiproliferative, anti-fibrotic, and anti-inflammatory effects [8,13].

### 1.2. Alternative Pathways of RAS

ACE inhibition does not entirely inhibit Ang II biosynthesis, as other proteases such as kallikrein, cathepsin G, heart chymase, and elastase-2 also converting Ang I to Ang II [14,15].

Moreover, Ang I and Ang II are alternatively cleaved into Ang (1-9) and Ang (1-7), respectively, by the action of the ACE2 enzyme, although Ang II is significantly preferred over Ang I as a substrate [16,17,18]. Ang (1-7) can also be formed from Ang (1-9) by ACE or directly from Ang I by endopeptidases such as neprilysin, or from Ang II by prolylcarboxypeptidase [19]. Increased ACE2 expression attenuates Ang II-induced cardiac hypertrophy while its reduced expression is associated with impaired cardiac contractility and hypertension development [20,21]. The biological effects of Ang (1-7) counteract many of the effects of Ang II, as it promotes vasodilation while inhibiting proliferation, hypertrophy, fibrosis, thrombosis, and rhythm disorders [19,22,23].

The beneficial effects of ACE inhibitors and AT1R blockers (ARB) are at least partially mediated through the increased accumulation of Ang (1-7). ACE inhibition leads to Ang I accumulation and its increased conversion to Ang (1-7) [24]. On the other hand, the blockade of AT1R by ARB leads to the accumulation of Ang II, which is subsequently converted to Ang (1-7) by ACE2 [24]. The target receptor of Ang (1-7) is the Mas receptor (MasR) while the aspartate decarboxylated product of Ang II alamandine binds to the MAS1-related G protein-coupled receptor D (MrgD), both taking part in the protective arm of RAS along with AT2R [25,26,27,28]. Additionally, the octapeptide angioprotectin is a more effective MasR agonist than Ang (1-7), which exerts vasodilatory action [29,30].

Angiotensin I can also be cleaved by aminopeptidase A and ACE into angiotensin III (Ang III) (Ang 2-8) and subsequent cleavage by aminopeptidase N leads to the formation of angiotensin IV (Ang IV) (Ang 3-8). Ang III can bind to both AT1R and AT2R, while Ang IV acts on AT4R, also known as insulin-regulated aminopeptidases (IRAP), located predominantly in the brain but also in the heart [8,31]. Ang IV is primarily associated with improved cognitive and memory functions, as it dose-dependently inhibits IRAP and induces the accumulation of neuropeptides [8,32].

N-acetyl-seryl-aspartyl-lysyl-proline (Ac-SDKP) and the adipokine apelin, other alternative substrates of ACE and ACE2 respectively, also join the cardioprotective arm of the RAS. Ac-SDKP administration exerts an anti-inflammatory, antiproliferation, antidifferentiation and anti-migration effect which in general attenuates cardiac fibrosis and Ac-SDKP augmentation by ACE inhibitors and partially mediates their pharmacological benefits [33]. Apelin binds to the GPCR APJ, exerting positive inotropic as well as cardioprotective effects [34,35,36].

To sum up, the RAS involves various molecules with opposing biological effects. One line is represented by vasoconstrictive, pro-proliferative and pro-inflammatory molecules, such as ACE, Ang II, and AT1R. The other line involves ACE2, Ang (1-7), AT2R, MasR, MrgD, and APJ, which exert effects at least partially counteracting the potentially harmful actions of the deleterious arm of the RAS (Figure 1).

Angiotensinogen produced from the liver is proteolytically activated to angiotensin I and II through the action of renin, ACE, chymase, kallikrein, cathepsin G, and esterase-2. Angiotensin II can bind to AT1R and AT2R to initiate the effect of the RAS, or can be further cleaved or modified into cardioprotective peptides, namely angiotensin IV, angiotensin (1-7), and alamandine, which exert their effect through AT4R, MasR and MrgD respectively. Angioprotectin, Ac-SKDP, Apelin have also emerged as new cardioprotective peptides of the RAS. Thus, the RAS can induce both pathological cardiac remodelling and cardioprotective effects, and the balance between the two determines the overall effect on the cardiovascular system. In chronic activation of the RAS, the cardiac remodelling arm is often stronger, as suggested by the therapeutic success of ACE inhibitors and AT1R blockers.

ACE, angiotensin converting enzyme; ACE2, angiotensin converting enzyme 2; APA, aminopeptidase A; APN, aminopeptidase N; PCP, prolylcarboxypeptidase; AD, aspartate decarboxylase; PRR, (pro)renin receptor; AT1R, angiotensin II receptor type 1; AT2R, angiotensin II receptor type 2; MasR, mas receptor; MrgD, MAS1-related G protein-coupled receptor D; Ac-SKDP, *N*-acetyl-seryl-aspartyl-lysyl-proline.

### 1.3. Local RAS

RAS components can be synthesized at a local tissue scale by the peptidases of the alternative pathways, a process known as the local RAS, which has been recognized to play an important role in tissue remodelling [37,38]. The local RAS is also activated by the binding of renin or prorenin to the (pro)renin receptor (PRR) located in various organs including the heart, and triggering a pro-fibrotic signalling cascade [39,40,41]. ACE is also distributed throughout the cardiovascular system and renal tissues, allowing the local synthesis of Ang II [37,42].

### 1.4. Intracellular or Intracrine RAS

The pro-remodelling arm of the RAS may also be mediated by the intracellular synthesis of renin and Ang II in cardiac fibroblasts [43,44,45]. High glucose levels and isoproterenol elevate intracellular renin synthesis, which is converted to Ang II through intracellular ACE, subsequently elevating pro-fibrotic transforming growth factor-β (TGF-β) and collagen-1 synthesis in neonatal rat ventricular fibroblasts [46]. High-glucose-induced intracellular Ang II synthesis as well as superoxide production and cardiac fibrosis were only fully blocked by aliskiren (direct renin inhibitor) but not by ARB or ACE inhibitors [47].

## 2. Angiotensin II Signalling and Cardiac Remodelling

Injurious insults trigger compensatory reactions in cardiomyocytes, cardiac fibroblasts, and inflammatory immune cells, leading to morphological and functional changes involving both parenchymal and interstitial tissues, known as remodelling [48]. Pathological cardiac hypertrophy with excessive fibrosis may serve as a typical example [49,50].

RAS-induced hemodynamic alterations as well as its direct trophic effect can contribute to developing or maintaining cardiac remodelling. Cardiomyocytes express both AT1R and AT2R, while cardiac fibroblasts exclusively express AT1R under normal circumstances, and AT2R in certain pathologies [51,52,53]. Inflammatory cells can also be activated by AT1R stimulation [54]. The physical stretch of cardiomyocytes has been suggested to stimulate local Ang II production, which subsequently activates p53, inducing apoptosis [55]. ARB losartan was able to attenuate stretch-induced apoptosis, demonstrating the importance of Ang II signalling [55].

The binding of Ang II to AT1R leads to the initiation of a wide spectrum of intracellular signalling pathways; this includes the protein kinase C (PKC), the mitogen-activated protein kinase (MAPK), a variety of tyrosine kinases, scaffold proteins, the transactivation of growth factor receptors, NADPH oxidase (NOX), and ROS signalling and G protein-independent pathways, including the β-arrestin and JAK/STAT [56,57]. These signalling pathways eventually activate hypertrophic/fibrotic transcription factors or proteins, as well as other signalling pathways such as the pro-hypertrophic calcineurin/nuclear factor of the activated T-cell (NFAT) pathway or the pro-fibrotic TGF-β pathway [58,59].

## 3. microRNAs in the Regulation of RAS-Induced Cardiac Remodelling

MicroRNAs (miRNAs) are noncoding regulatory RNAs of ~22 nucleotides in length, involved in the silencing of genetic transcripts by binding to 3′ UTR of its target genes in eukaryotic cells [60,61]. A single miRNA can target a multitude of genes, while a single gene can be the target of a multitude of miRNAs [62]. While under physiological conditions, the main functions of miRNAs are considered to be the fine-tuning of gene expression; its silencing effect becomes more prominent under pathological conditions [63]. Since its relatively recent discovery, miRNA involvement has been implicated in the majority of pathological states, including RAS-induced cardiac remodelling [64,65,66,67,68]. As the following sections illustrate, miRNAs seem to directly target the RAS and Ang II signalling at all levels, including enzymes that are involved in the generation of Ang II (renin, ACE), its receptors (AT1R, AT2R), and compounds of Ang II signalling pathways.

### 3.1. miRNAs Involved in Ang II-Induced Cardiac Hypertrophy

Ang II-induced hypertrophy in cardiomyocytes is primarily mediated by AT1R through the canonical pathway, the epidermal growth factor receptor (EGFR) transactivation pathway, and the inflammatory pathway (reviewed in [69]).

The canonical pathway is initiated through the activation of the Gα13 protein, which in turn activates the calcineurin/NFAT, ERK1/2 pathways and RhoA [70,71]. RhoA is a small GTPase that induces further downstream activation of myocardin-related transcription factors, leading to the expression of both hypertrophic and fibrotic genes [72].

EGFR transactivation is initiated by the Ang II-induced increase in the intracellular Ca^2+^ and ROS, followed by the Src phosphorylation of ADAM metallopeptidase domain 17 (ADAM17). ADAM17 catalyses the shedding of the heparin-binding EGF-like growth factor (HB-EGF), the ligand for EGFR, resulting in a pro-hypertrophic MAPK, and phosphoinositide 3-kinase/protein kinase B (PI3K/AKT) signalling [73].

Lastly, additional pro-hypertrophic growth-promoting signals can also be obtained from the AT1R-induced inflammatory pathway, which is mediated by the release of a variety of inflammatory cytokines including IL-1, IL-6, and TNF-α, along with NOX2 and NOX4 activation and subsequent ROS generation [74,75,76,77,78,79,80,81,82,83].

Furthermore, AT1R-induced mitochondrial NOX4 activation leads to elevated mitochondrial ROS, which induces hypertrophy-promoting autophagy [84]. Indeed, reduced ROS due to the cardioprotective Ang (1-7)/MasR axis reduces cardiac autophagy as well as hypertrophy, while the opposite action was observed with the deletion of the anti-inflammatory cytokine IL-10 [85,86].

The miRNA expression in Ang II-mediated cardiac hypertrophy is known to be deleterious, and now a growing list of miRNAs is associated with cardiac hypertrophy. Specifically, miR-155, miR-208, miR-132, miR-212, miR-21, miR-410, miR-495, miR-19a, miR-19b, and miR-20b have been associated with promoting hypertrophy, while miR-21-3p, miR-26a, miR-16, miR-98, miR-30a, 34a, miR-133a, miR-1, miR-99a, miR-101, and miR-129-3p have been associated with the prevention or reversal of cardiac hypertrophic growth; however, conflicting evidence also exists and is addressed in the following sections (Figure 2, Table 1).

AT1R triggers a variety of signalling pathways that can be divided into the canonical pathway depicted at the centre of the figure: the EGFR transactivation pathway depicted to the left and the inflammatory pathway depicted on the right (based on [69]). Anti-hypertrophic miRNAs are depicted in blue, while pro-hypertrophic ones are depicted in red.

AT1R, angiotensin II receptor type 1; PLC, phospholipase C; IP3, inositol triphosphate; PKC, protein kinase C; CaM, calmodulin; NFAT, nuclear factor of activated T-cell; CYTOR, cytoskeleton regulator RNA; BIC, B-cell integration cluster; IKKi, I-kappa-B kinase epsilon; NF-κB, nuclear factor kappa-light-chain-enhancer of activated B cells; NOX, NADPH oxidase; ROS, reactive oxygen species; ADAM17, ADAM metallopeptidase domain 17; HB-EGF, heparin-binding EGF-like growth factor; EGFR, epidermal growth factor receptor; MAPKKK, mitogen-activated protein kinase kinase kinase; MAPKK, mitogen-activated protein kinase kinase; MAPK, mitogen-activated protein kinase; ERK1/2, extracellular signal-regulated kinase 1/2; JNK, c-Jun N-terminal kinase; PI3K, phosphoinositide 3-kinase; AKT, protein kinase B; mTOR, mechanistic target of rapamycin; p70S6K, ribosomal protein S6 kinase 1; FoxO3, forkhead box protein O3; MuRF-1, muscle RING-finger protein-1; ATG9A, autophagy-related protein 9A; SORBS2, SH3 domain-containing protein 2; PDLIM5, PDZ and LIM domain 5; PKIA, protein kinase A inhibitor.

#### 3.1.1. Pro-Hypertrophic miRNAs

##### miR-155

miR-155 mediates inflammatory processes that promote cardiac hypertrophy secondary to Ang II stimulation. It is generated from the B-cell integration cluster (*BIC*) gene, whose promoter is under the regulation of the activator protein-1 (AP-1) and the nuclear factor κ-light-chain-enhancer of activated B cells (NF-κB) [87,88]. Normally, the level of miR-155 is low in healthy individuals, and its upregulation is generally associated with the activation of an innate immune response [89]. In mice macrophages, loss of miR-155 significantly attenuated cardiac inflammation and remodelling [90]. The pro-hypertrophic effect of miR-155 is at least partially mediated by its directly targeting suppressor of cytokine signalling (Socs1), as the Socs1 knockdown largely re-established the hypertrophy-promoting ability of miR-155 knockout macrophages [90]. Another study indicated that the loss of miR-155 in mice hearts also prevented transverse aortic constriction (TAC) and calcineurin activation-induced cardiac hypertrophy, with Jumonji, the AT-rich interactive domain 2 (Jarid2) identified as another potential target of action [91]. Yuan et al. showed that miR-155 reduces I-kappa-B kinase epsilon (IKKi) protein levels and thus activates the pro-hypertrophic NF-κB pathway [92]. The long noncoding RNA (lncRNA) cytoskeleton regulator RNA (CYTOR) was identified as an miR-155 sponge, and as the CYTOR knockdown accelerated Ang II-induced cardiac hypertrophy, it was suggested that the CYTOR/miR-155 axis plays an important modulatory role in Ang II-induced pro-hypertrophic NF-κB signalling [92]. Furthermore, miR-155 was recently found to be packaged into exosomes and released from Ang II-induced hypertrophic cardiomyocytes, triggering inflammatory cytokine IL-6/8 release from macrophages [93].

As opposed to other reports supporting miR-155 in promoting cardiac remodelling, Yang et al. found that significantly increased miR-155 expression in Ang II-treated H9c2 rat cardiomyoblast cells led to reduced cell surface area and attenuated AT1R, ANP, β-MHC, and calcineurin/NFATc4 mRNA levels and intracellular free Ca^2+^ levels, strongly suggesting that miR-155 attenuates Ang II-induced hypertrophy through downregulating AT1R and its downstream Ca^2+^ signalling [94].

Furthermore, miR-155 could modulate the effect of Ang II on vascular inflammation and remodelling through the direct inhibition of AT1R, at least in human umbilical vein endothelial cells (HUVECs) [95,96]. Another direct target also verified in HUVECs is endothelial nitric oxide synthase (eNOS), which can at least partially mediate the effect of tumour necrosis factor-α (TNF-α) in endothelial cells [97]. TNF-α also triggers endothelial cells to release microparticles, while pre-miR-155 and miR-155 microparticles were found to be significantly increased in response to TNF-α [98]. In vascular adventitial fibroblasts, by targeting AT1R, miR-155 attenuated Ang II-induced ERK1/2 activation, and α-SMA deposition, while TGF-β1 administration largely abolished this effect [99]. miR-155 also opposes Ang II-induced dose- and time-dependent increases in the viability and proliferation of vascular smooth-muscle cells (VSMCs) [100].

In summary, miR-155 demonstrates a hypertrophic effect through targeting Soc1, Jarid2, and IKKi, thereby augmenting inflammatory NF-κB signalling downstream from Ang II stimulation. On the contrary, it has also been reported to antagonize against hypertrophy through reduced AT1R and Ca^2+^ signalling. Its direct inhibitory action on AT1R and eNOS could also play an important role in modulating Ang II-induced vascular inflammation and remodelling.

##### miR-208

The miR-208 family is specific to the heart, and their levels are significantly altered in heart failure [101]. The miR-208 family consists of miR-208a and miR-208b, co-expressed by the fast myosin *Myh-6* gene and the slow myosin *Myh-7* gene, respectively, as they are encoded in the intron portion [102]. Additionally, the miR-208 family shows that its expression is specifically related to a particular part of the myocardium. While miR-208a is highly localized in the left atrium, miR-208b is preferentially expressed in the left ventricle [103]. Transgenic miR-208a overexpression leads to enhanced *Myh7* expression and cardiac hypertrophy in mice via inhibition of target proteins, namely the two negative regulators of hypertrophic growth, thyroid hormone-associated protein 1, and myostatin [104]. miR-208a can also induce arrhythmia, while miR-208a knockout mice showed impaired cardiac conduction as well as impaired expression of GATA4, connexin 40, and homeodomain-only protein [104]. miR-208a silencing through the systemic administration of antisense oligonucleotide has been shown to be cardioprotective, inhibiting cardiac remodelling and pathological myosin switching, thus indicating its potential as a therapeutic target [105]. miR-208 is considered to also be involved in apoptosis, although the data are rather conflicting [106,107]. Importantly, Ang II signalling has been implicated in the modulation of miR-208 levels and cardiac remodelling. Ang II reduced miR-208 and increased apoptosis in the H9c2 rat cardiomyoblast cell line [108]. While miR-208 mimics reduce the ratio of apoptotic cells, miR-208 inhibitors increase it [108]. Meanwhile, it has also been reported that Ang II-induced cardiomyocyte apoptosis can be exacerbated by miR-208a through the inhibition of Nemo-like kinase (NLK) [109].

In short, the miR-208 family promotes hypertrophy and arrhythmia while also contributing to apoptosis. Cardiomyocytes treated with Ang II attenuates miR-208 expression and promotes apoptosis, at least partially through NLK inhibition.

##### miR-132/212

miR-132/212 has been widely studied in diseases of the nervous system, but evidence suggests its involvement in RAS-induced cardiac remodelling, especially in the context of renal injury. Cardiac miR-132/212 is elevated after subtotal nephrectomy in rats, and the level was attenuated by ramipril [110]. A global miRNA array analysis of AT1R-overexpressing human embryonic kidney 293 (HEK293N) cells has also revealed that miR-132 and 212, along with miR-29b and miR-129-3p, were upregulated after Ang II stimulation, through Gαq/11 and ERK1/2 activation [111]. miR-132/212 has been found to be pro-hypertrophic and anti-autophagic, and its elevation is considered a necessary and sufficient factor for inducing cardiac hypertrophy by targeting the forkhead box O3 (FoxO3) transcription factor gene, thus activating calcineurin/NFAT signalling and leading to the hypertrophy of cardiomyocytes [112]. Indeed, in a rat model, miR-132/212 is upregulated in the heart, aortic wall, and kidney in Ang II-induced cardiac hypertension and hypertrophy [113]. Surplus arterial tissue from patients after coronary bypass surgery revealed that ARB reduced the level of miR-132/212, while β-blockers did not [113]. In silico analysis and in vitro miRNA arrays have identified a myriad of genes that could be targeted by miR-132/212, including seven genes involved in Ang II signalling, namely AT1R, AC, JAK2, PKC, c-JUN, SOD2, and EGR1, suggesting their potential modulatory role in Ang II signalling [114]. Importantly, miR-132/212 could be a promising therapeutic target, as the inhibition of miR-132 with antagomir was shown to reduce cardiac morbidity through the attenuation of hypertrophy and heart failure at least in mice [112]. Recently, an optimized, synthetic locked nucleic acid antisense oligonucleotide inhibitor (antimiR-132) has been tested for its therapeutic potential in a mouse model as well as in a pig model of heart failure, and antimiR-132 has demonstrated not only its promising cardioprotective effect but its safety, tolerability, favourable pharmacokinetic profile, and a strong PK/PD relationship [115].

In summary, miR-132/212 is elevated by Ang II stimulation and promotes hypertrophy through directly inhibiting FoxO3, thereby activating calcineurin/NFAT signalling. The suppression of their levels through ARBs or antagomirs could be beneficial as they lower cardiac hypertrophy.

##### miR-21

miR-21 has been found to be significantly upregulated in both mice and human models of cardiac hypertrophy or heart failure in multiple expression profiling studies, and its elevated level of expression is associated with the acceleration of cardiac remodelling [116,117]. On the other hand, miR-21 knockdown also showed a significant reduction of cardiomyocyte hypertrophy [116].

##### miR-410, miR-495

In chronic Ang II stimulation, along with other models such as myocardial infarction and muscular dystrophy-associated cardiomyopathies, miR-410 and miR-495 are elevated and promote cardiac hypertrophy, whereas miR-410/495 inhibition attenuated hypertrophic growth [118].

##### miR-19a/b

Conflicting reports exist regarding the function of miR-19a/b in cardiac hypertrophy. Song et al. provided evidence for its pro-hypertrophic role, as anti-hypertrophic genes atrogin-1 and muscle RING-finger protein-1 (MuRF-1) are directly targeted by miR-19a/b, leading to augmented pro-hypertrophic calcineurin/NFAT signalling as well as PKC signalling [119]. The inhibitors of the calcineurin/NFAT pathway (cyclosporin A) and the PKC pathway (GF10923X) largely abolished the hypertrophic effects of miR-19b [119]. miR-19b was also found to reduce apoptosis and improve cardiomyocyte survival through targeting the proapoptotic Bcl-2-interacting mediator of cell death (Bim) and phosphatase and tensin homolog (PTEN) [119,120]. Meanwhile, another study found that Ang II significantly reduced miR-19b expression in cultured cardiomyocytes, while significantly increasing the pro-hypertrophic connective tissue growth factor (CTGF) and that both could be mitigated through ARB telmisartan treatment [121]. Since miR-19b overexpression reduced CTGF levels and miR-19b transfection prevented the Ang II-induced elevation of CTGF, it was concluded that Ang II-induced CTGF overexpression is mediated by attenuated miR-19b levels [121]. Liu et al. found a rather anti-hypertrophic effect of miR-19a/b-3p in Ang II-induced cardiac hypertrophy by directly targeting phosphodiesterase 5A (PDE5A) [122]. miR-19a/b-3p were downregulated in pressure-overload-induced cardiac hypertrophy, while overexpression with transgenic mice prevented such hypertrophic changes following Ang II infusion [122].

In conclusion, miR-19a/b may promote cardiac hypertrophy through targeting atrogin and MuRF-1 or they may inhibit apoptosis through targeting Bim and PTEN. However, they might also oppose hypertrophy through the reduction of CTGF and PDE5A, and Ang II-induced attenuated levels of miR-19a/b could mediate cardiac hypertrophy.

##### miR-20b

miR-20b was found to be elevated in an Ang II-induced cardiac hypertrophy model using neonatal rat ventricular cardiomyocytes (NRVCs) and miR-20 inhibition using antisense inhibitors reversed the hypertrophy [123]. A subsequent luciferase reporter assay identified mitofusin 2 (Mfn2) as the direct target of inhibition, and the authors suggested the dysregulated mitochondrial Ca^2+^ buffering of the potential pathogenic mechanism [120]. miR-20b-5p also directly targets small mothers against decapentaplegic homolog 7 (SMAD7), which has been shown to counter Ang II-induced hypertensive cardiac remodelling. Increased miR-20b-5p levels reduce SMAD7, thus worsening cardiac function in a rat model of ischemic-reperfusion injury [124,125].

#### 3.1.2. Anti-Hypertrophic miRNAs

##### miR-21-3p, miR-26a

miR-21-3p has been reported to be involved in Ang II-induced cardiac hypertrophy, although evidence suggests its role both in inhibiting and promoting hypertrophy.

The anti-hypertrophic effect was shown to be mediated by miR-21-3p directly targeting histone deacetylase 8 (HDAC8) related to PTEN, which augments hypertrophic signalling through elevating phospho-AKT and phospho-Gsk3β levels [126]. miR-26a was also investigated through transverse aortic constriction (TAC) in rats and in Ang II-treated cardiomyocytes and was found to directly target pro-hypertrophic GATA4 to exert an anti-hypertrophic effect [127].

As a pro-hypertrophic agent, miR-21-3p can be expressed in cardiac fibroblasts, and rather than being degraded inside the cell, it can be packaged into exosomes and delivered to cardiomyocytes in a paracrine fashion [128]. The SH3 domain-containing protein 2 (*SORBS2*) and PDZ and LIM domain 5 (*PDLIM5*) have been identified as target genes in cardiomyocytes, and the silencing of these genes leads to hypertrophy [128].

##### miR-16, miR-98

Cell-cycle regulators cyclins have been implicated in the pathogenic process of Ang II-induced myocardial hypertrophy. miR-16 suppresses hypertrophy by targeting pro-hypertrophic cyclin D1, D2, and E1, and its level is reduced in cardiac hypertrophy due to the expression of STAT3/c-Myc [129]. miR-98 is another anti-hypertrophic miRNA that targets cyclin D2, which is upregulated by Ang II [130].

##### miR-30a, miR-34a

Cardiac autophagy that promotes hypertrophy is also the target of regulation by miRNAs. The autophagy-related gene beclin-1 and ATG9A are targeted by miR-30a and miR-34a, respectively [131,132]. Both miR-30a and miR-34a are downregulated in cardiomyocytes in a rat model of cardiac hypertrophy, upregulating their autophagy-related target genes, resulting in enhanced autophagy and hypertrophy-related genes such as the atrial natriuretic peptide (ANP) and the β-myosin heavy chain (β-MHC) [131,132]. The overexpression of miR-30a or miR-34a attenuated both cardiac autophagy and hypertrophy [131,132].

##### miR-133a, miR-1

miR-133a and miR-1, along with miR-208a and miR-499, are miRNAs expressed specifically in the myocardial tissue in humans. Both miR-133a and miR-1 are important for early cardiogenesis, while miR-208a and miR-499 are more important for later periods of cardiogenesis [133]. miRNA-133 is considered to play an anti-hypertrophic and anti-fibrotic role in the human heart [134,135,136,137,138,139]. Its level is reduced in cardiac hypertrophy along with miR-1, and curiously, the antisense ‘decoy’ antagomir targeting miR-133a was sufficient to induce cardiac hypertrophy [136]. However, hypertension cannot be prevented by restoring miR-133a levels during pressure overload [138,140].

Several targets for miR-133 have been validated in cardiomyocytes and heart 293HK cells or Cos-1 cells or C2C12 cells, including SRF, cyclin D2, caspase 9, CTGF, RhoA, Cdc42, Nel-A/WHSC2, collagen 1α1 and NFATc4 [135,141,142,143,144,145,146]. Additionally, a bioinformatics study followed by a luciferase reporter assay revealed that the overexpression of miR-133a led to reduced levels of angiotensinogen and AR1R miRNA. In congestive heart failure (CHF) in rats, changes in angiotensinogen and AT1R expression have been described in the periventricular nucleus (PVN) [147]. Thus, miR-133a is now implicated to play a modulatory role in the RAS-induced increase in sympathetic tone. Indeed, the viral transduction of miR-133a in the PVN of these CHF rats led to a significant reduction in angiotensin and AT1R levels as well as reduced basal renal sympathetic nerve activity [147]. Yet another target of miR-133a is collagen 1α1 (Col1α1), verified by a luciferase assay and miR-133a binding site-mutated Col1α1 mRNA [145]. Notably, the particular mechanism to the downregulation of miR-133a seems to be mediated by the AMPK/ERK1/2 pathway, as the ERK inhibitor PD98059 abolishes Ang II-induced reduction in miR-133a levels [148]. This regulatory pathway was also shown to participate in the anti-hypertrophic effect of adiponectin [148]. Furthermore, miR-133a and miR-1, along with miR-208, have been associated with mediating the effects of the ACE2/apelin pathway. Cardiac hypertrophy and dysfunctional cardiac contractility induced by a high-fat diet were found to be reversed by apelin treatment through the elevation of miR-133a, miR-1, and miR-208 [149]. Its suppressive role in cardiac remodelling could potentially make miR-133a a useful therapeutic agent, and some studies have already reported the beneficial effects of miR-133a overexpression in mice models of diabetic cardiomyopathy [150,151].

miR-1 is also considered to be anti-hypertrophic through NFATc3 targeting of the calcineurin/NFAT pathway in cardiomyocytes [152]. Its level is decreased in patients with hypertrophic hearts as well as in rat cardiomyocytes treated with pro-hypertrophic stimuli, and its suppression was sufficient to trigger cardiac hypertrophy [152]. Meanwhile, miR-1 overexpression inhibited such hypertrophic change, suggesting the therapeutic potential of targeting the calcineurin/NFAT pathway through miR-1 [152]. Another study also suggested that miR-1 might target the mitochondrial calcium uniporter (MCU), a critical component of intracellular Ca^2+^ homeostasis and stress adaptation in cardiomyocytes [153]. Biopsies from human patients with cardiac hypertrophy showed reduced miR-1 expression in the heart was associated with enhanced MCU protein content [153].

The cytoskeleton regulatory protein twinfilin-1 is another verified direct target of miR-1 which, upon miR-1 downregulation, induces a pro-hypertrophic effect in cardiomyocytes [154]. In a rat model of cardiac hypertrophy, a significant reduction in miR-1 expression level was found to elevate another pro-hypertrophic direct target cyclin D kinase 6 (CDK6) [155]. Lastly, the insulin-like growth factor-1 (IGF-1) and its receptor are also both targets of miR-1 and IGF-1 levels correlated with the extent of cardiac hypertrophy [156]. IGF-1 also provides a feedback regulation of miR-1 expression through the forkhead box O3a (FoxO3a) transcription factor [156].

Taken together, miR-133a targets a variety of pro-hypertrophic genes, namely SRF, cyclin D2, CTGF, RhoA, NFATc4, angiotensinogen, and AT1R, thus opposing cardiac hypertrophy. Its reduction in CHF is associated with elevated angiotensinogen and subsequent Ang II levels in PVN, leading to an increased sympathetic tone. Apelin treatment exerts a cardioprotective role by elevating miR-133a. miR-1 is also anti-hypertrophic by targeting NFATc3 and MCU, thus attenuating pro-hypertrophic calcineurin/NFAT and Ca^2+^ signalling. miR-1 also targets other pro-hypertrophic proteins, namely twinfilin-1, CDK6, IGF-1 and its receptor.

##### miR-99a, miR-101

miR-99a has been found to directly target the mechanistic target of rapamycin (mTOR) in Ang II treated cardiomyocytes, and its overexpression attenuated a mice model of hypertrophy induced by TAC [157]. Another miRNA that attenuates Ang II-induced cardiac hypertrophy is miR-101, which targets Rab1a [158].

##### miR-129-3p

The cardioprotective effect of Ang (1-9) has also been identified to be at least partially mediated by miR-129-3p, which attenuates protein kinase A inhibitor (PKIA) transcript levels, resulting in protein kinase A (PKA) signalling activation. PKA signalling is required for mitochondrial fusion and calcium homeostasis regulation induced by Ang (1-9) which antagonizes against cardiac hypertrophy [159].

### 3.2. miRNAs in Ang II-Induced Cardiac Fibrosis

Ang II-induced ROS production plays a central role in the pro-fibrotic downstream signalling to AT1R. The primary sources of ROS are likely to be NOX2 and NOX4, which are the dominant isoforms in the heart, as well as the mitochondria [83,160,161]. Specifically, the mitochondrially located pattern recognition receptor nucleotide-binding domain and leucine-rich repeat-containing PYD-3 (NLRP3) represent a major source of Ang II- and TGF-β-induced ROS in mice cardiac fibroblasts, promoting fibrosis [162].

Although a substantial amount of ROS can oxidize biomolecules and directly contribute to cell damage and death, the transient increase in ROS triggered by growth factors, hormones, or various inflammatory cytokines can contribute to redox signalling and activate the stress response pathway [163]. In Ang II-treated cardiomyocytes and cardiac fibroblasts, ROS can activate a variety of signalling cascades, including AKT/mTOR, ERK1/2, p38, NF-κB, RhoA, matrix metalloproteinase-2 (MMP-2), and the TGF-β pathway that lead to the expression of fibrotic genes, such as extracellular matrix (ECM) proteins, MMP-9, α-smooth muscle actin (α-SMA), and periostin [83,164,165,166,167,168,169]. It is worth noting that the activation of each pathway possesses some redundancy, and intracellular Ca^2+^ and PKC-*δ* can also activate such pro-fibrotic ERK1/2 pathway, while ERK1/2 or p38 themselves can also augment the TGF-β pathway [170,171,172]. RhoA is activated downstream from NOX-dependent ROS generation, and by Gα13 and Gq/11 downstream from AT1R [72,169,173,174]. RhoA subsequently increases SRF activity and CTGF secretion, contributing to increased ECM deposits [174,175]. Moreover, in cardiac fibroblasts, NOX4-induced ROS downstream from AT1R leads to pro-inflammatory IL-18 expression, as well as MMP-9, lipoxygenase (LOX), and collagen expression, which in general favours fibroblast migration and ECM deposition [176]. ROS contributes to the TGF-β pathway, increasing the activity of both canonical SMAD-dependent pathways and noncanonical signalling through p38 and ERK1/2 signalling and SRF/CREB transcription factors [69,177]. Furthermore, as with hypertrophy, the transactivation of EGFR through ROS in cardiomyocytes also contributes to fibrosis via MAPK/ERK or PI3K/AKT pathways [178,179]. In addition to these redox-sensitive signalling proteins, ROS induces the cytoplasmic translocation of the RNA-binding protein HuR, which is followed by an augmented TGF-β pathway [180].

The multifactorial cytokine TGF-β family is a key regulator of pro-fibrotic genes, favouring ECM deposition in many organs including the heart [181]. It consists of three isoforms, of which TGF-β1 is mainly implicated in cardiac remodelling [182,183,184,185]. Extensive evidence suggests that Ang II can induce TGF-β1 expression through AT1R signalling, leading to auto-/para-crine responses in cardiomyocytes and cardiac fibroblasts [59]. In fact, the hypertrophic and fibrotic effect of Ang II is completely absent if TGF-β1 is knocked out [186]. In ventricular myocytes, upon Ang II binding to the Gq-coupled receptor AT1R, the NOX-induced sequential activation of PKC, MAPK, and AP-1 triggers TGF-β1 expression [187]. In VSMCs and primary rat cardiac fibroblasts, Ang II can also activate the SMAD pathway independent of TGF-β [188,189,190,191,192].

TGF-β1 signalling is initiated upon TGF-β binding to and assembly of TGF-β receptor type I and type II serine/threonine kinases (TGFBR1, TGFBR2), followed by the canonical SMAD pathway or the noncanonical non-SMAD pathways including Rho-like GTPase pathways, MAPK pathways, and PI3K/AKT pathways [193,194]. SMAD proteins can be categorized into three groups based on their functions [195]. The receptor-regulated R-SMADs (SMAD2, SMAD3) are substrates for TGF-β family receptors. The common Co-SMAD (SMAD4) functions as a common binding target for the R-SMADs and the complex can subsequently be transported into the nucleus. The inhibitory I-SMADs (SMAD6, SMAD7) compete with SMAD4 for R-SMAD binding, and upon binding, they inhibit R-SMAD function. The TGF-β/SMAD pathway has been implicated in inflammation and tissue injury-induced cardiac fibrosis [185]. Specifically, SMAD3 is particularly important for its pro-fibrotic effect, while inhibitory SMAD7 antagonizes it [196,197]. Curiously, SMAD2 deletion would increase fibrosis by upregulating SMAD3 [198]. Non-SMAD pathways in cardiac fibrosis have yet to be systematically studied, but at least TGF-β-activated kinase 1 (TAK1) has been suggested to play a role in the process through its action on MKKs and the resulting phosphorylation of JNK and p38 [54,199,200]. The level of TAK1 activity was found to be elevated 7 days after increased mechanical load with aortic banding, while an activating mutation induces cardiac hypertrophy and interstitial fibrosis [199].

Several studies have demonstrated the involvement of miRNAs in regulating myocardial fibrosis in the settings of myocardial ischemia or mechanical overload. Namely, miR-21, miR-433, miR-503, miR-34a, and miR-155 have been shown to favour fibrogenesis, being pro-fibrotic miRNAs, while miR-26a, miR133a, miR-19a/b-3p, miR-29b, miR-22, and miR-let-7i have been found to be anti-fibrotic miRNAs (Figure 3, Table 1).

Ang II induces ROS production through NOX2, NOX4, and NLRP3. Intracellular ROS serves an important signalling function by activating various pro-fibrotic pathways (based on [69]). TGF-β pathway is another indispensable pathway for Ang II-induced cardiac fibrosis, and it can also be activated through elevated ROS. The EGFR transactivation pathway is more important in cardiomyocytes. Anti-fibrotic miRNAs are depicted in blue, while pro-fibrotic miRNAs are depicted in red.

AT1R, angiotensin II receptor type 1; NOX, NADPH oxidase; NLRP3, nucleotide-binding domain and leucine-rich repeat-containing PYD-3; ROS, reactive oxygen species; EGFR, epidermal growth factor receptor; NF-κB, nuclear factor kappa-light-chain-enhancer of activated B cells; AP-1, activator protein-1; IL-18, interleukin-18; STAT3, signal transducer and activator of transcription 3; PTEN, phosphatase and tensin homolog; Spry1, sprouty homolog 1; RECK, reversion-inducing cysteine-rich protein with Kazal motifs; MMP-2, matrix metalloproteinase-2; MAPK, mitogen-activated protein kinase; ERK1/2, extracellular signal-regulated kinase 1/2; JNK, c-Jun N-terminal kinase; PI3K, phosphoinositide 3-kinase; AKT, protein kinase B; mTOR, mechanistic target of rapamycin; HuR, human antigen R; TGF-β, transforming growth factor-β; SMAD, small mothers against decapentaplegic homolog; TRAF6, TNF receptor associated factor-6; TAK1, TGF-β-activated kinase 1; MKK, mitogen-activated protein kinase kinase; SRF, serum response factor; CREB, cAMP response element-binding protein.

#### 3.2.1. Pro-Fibrotic miRNAs

##### miR-21

miR-21 shows a close relationship with cardiac fibrosis, and its levels are selectively enhanced in myocardial infarction and heart failure [201,202]. miR-21 exerts its action by augmenting the MAPK pathways and PI3K/AKT pathways through the direct inhibition of sprouty homolog 1 (Spry1), an inhibitor of the MAPK pathway and phosphatase and tensin homolog (PTEN), an inhibitor of PI3K [201,203,204]. The Ang II-induced activation of Rac1 causes the CTGF- and lysyl oxidase-mediated elevation of miR-21, which in turn inhibits Spry1, leading to enhanced pro-fibrotic MAPK signalling contributing to atrial fibrosis [205]. What is more, AP-1 and STAT3 activation downstream from the AT1R/NOX/ERK1/2 axis increases miR-21, which directly suppresses PTEN, SMAD7, Spry1, as well as MMP regulator reversion-inducing cysteine-rich protein with Kazal motifs (RECK), leading to increased MMP-2 expression and favouring fibroblast migration, survival, and collagen deposition [202,206,207].

miR-21 is also implicated in TGF-β signalling, in both the SMAD and the non-SMAD pathways. In cardiac fibroblasts, after TGF-β receptor stimulation, the pro-fibrotic SMAD3 inhibits SMAD7 through the induction and processing of pre-miR-21 to mature miR-21, which directly targets SMAD7 [208,209,210,211]. miR-21 also directly targets TGFBR2, PTEN, and Spry1, suggesting their active role in the noncanonical PI3K/AKT pathways and MAPK pathways of TGF-β signalling as well [201,202,204,212]. TGF-β-induced endothelial/epithelial mesenchymal transition (EndoMT/EMT) is also partially mediated by miR-21 via its direct inhibition of PTEN and the activation of the PI3K/AKT pathway [183,213].

Taken together, miR-21 expression is directly elevated in response to Ang II stimulation, and pro-fibrotic MAPK, PI3K/AKT and TGF-β pathways are augmented by the action of miR-21 targeting the pathway inhibitors, namely Spry1, PTEN, and SMAD7. Indeed, the anti-fibrotic effect of Ang (1-7) is mediated by the inhibition of miR-21 expression secondary to Ang II stimulation; this was reported in lung fibroblasts, and if a similar action exists in cardiac fibroblasts it has yet to be studied [214].

##### miR-433

miR-433 is another pro-fibrotic miRNA whose knockdown suppressed myofibroblasts transdifferentiation and fibrosis after TGF-β or Ang II treatment, while overexpression of miR-433 had the opposite effect and promoted fibrosis [215]. The same authors identified JNK1 and AZIN1 as the key target genes that mediate the fibrotic effect, as reduced JNK1 leads to ERK and p38 activation resulting in the activation of pro-fibrotic SMAD3, while reduced AZIN1 activates TGF-β1.

##### miR-503

miR-503 has been shown to promote Ang II-induced cardiac fibrosis by targeting Apelin-13, a cardioprotective adipokine peptide that antagonizes against Ang II-induced CTGF and TGF-β activations in neonatal cardiac fibroblasts [216]. Targeting Apelin by miR-503 has also been associated with other cardiovascular conditions, including the inhibition of hypoxia-induced endothelial progenitor cell proliferation, migration, and angiogenesis, as well as high-glucose-induced microvascular cell injury through enhanced inflammation and oxidative stress [217,218].

##### miR-34a

miR-34a has been suggested both as a potential biomarker and a therapeutic target for cardiac remodelling, following several conditions such as acute myocardial infarction and anthracycline-induced cardiotoxicity, but it has not been directly associated with fibrosis secondary to Ang II stimulation [219,220,221,222,223]. However, Ang II pro-fibrotic signalling is crucially mediated by TGF-β1, which has a strong association with miR-34a. TGF-β1 upregulated miR-34a levels in cardiac fibroblasts, and the attenuation of miR-34a levels was associated with reduced fibrotic activity [224]. The pro-fibrotic effect is at least in part modulated via direct interaction with SMAD4, as overexpression of miR-34a leads to increased SMAD4, while the inhibition of miR-34a leads to the opposite; however, this result is rather contradictory to the inhibitory nature of miRNAs [224].

##### miR-155

As mentioned above, the inflammatory process is crucial to the Ang II-induced fibro-genic process, and miR-155 expression in cardiac fibroblasts will favour such pro-fibrotic inflammation [225]. miR-155-deleted mice were resistant to Ang II-induced inflammatory responses and myocardial fibrosis compared to wild-type mice [225]. miR-155 overexpression in vitro favoured fibroblasts to transdifferentiate into more fibrotic myofibroblasts, which was abolished with miR-155 silencing [225]. Evidence also indicates that the pro-fibrotic action of miR-155 can be mediated by augmenting TGF-β1 signalling [226,227].

#### 3.2.2. Anti-Fibrotic miRNAs

##### miR-26a

In addition to its anti-hypertrophic effect, miR-26a is implicated in the inhibition of Ang II-induced cardiac fibrosis through the direct targeting of CTGF and collagen I genes [228]. miR-26a forms a negative feedback loop with NF-κB, where miR-26a overexpression reduces NF-κB activity, while NF-κB inhibition also restored miR-26a level in cardiac fibroblasts [228]. Cardiac-specific IκBa triple mutant transgenic mice with impaired NF-κB activation were shown to be relatively resistant against TAC-induced cardiac fibrosis compared to wild type mice thanks to miR-26a expression and attenuated CTGF and collagen I expression [228].

##### miR-133a

The direct targets of miR-133a include various pro-fibrotic factors, namely RhoA, SRF, CTGF and collagen 1α1 [135,141,144,145]. Notably, myocardial fibrosis in Ang II-induced hypertension attenuated miR-133a levels and elevated collagen 1α1, while the administration of ARB irbesartan eliminated such effect [145]. This finding indicates the potential participation of miR-133a in Ang II signalling modulation. As mentioned above, Ang II-induced ROS generation promotes RhoA activation and subsequent SRF activation and CTGF release, aiding in the fibrotic process [174]. miR-133a is involved in the regulation of this RhoA/SRF/CTGF axis of AT1R action, as SRF overexpression significantly attenuates miR-133a levels, and miR-133a represses SRF and CTGF expression [229].

##### miR-19a/b-3p

In addition to its pro-hypertrophic action, miR-19a/b-3p has been suggested to attenuate cardiac fibrosis. miR-19a/b-3p expressions are found to be low in patients with heart failure, and their direct inhibition of TGFBR2 has been suggested to reduce autophagy-related fibrosis [230]. Additionally, miR-19b-3p directly targets the Co-SMAD or SMAD4 needed for SMAD2, 3 to translocate into the nucleus [231,232]. Although these studies regarding fibrosis were not carried out in Ang II-induced models, since Ang II-induced cardiac fibrosis is also mediated by TGF-β signalling, it is possible that miR-19a/b-3p could be playing a role in that context.

##### miR-29b

miR-29b is widely considered to be anti-fibrotic in a variety of tissue fibrosis, as it binds to the 3’-UTR of more than 11 of the 20 collagen genes along with fibrillin and elastin, thus downregulating them against TGF-β stimulation [233,234,235]. Its overexpression both in vitro and in vivo inhibits Ang II-induced cardiac fibrosis while in vitro knockdown promotes fibrosis [236]. miR-29a-c levels are attenuated via the inhibition of TGF-β/SMAD3 signalling in both cultured cardiac fibroblasts and Ang II-induced cardiac fibrosis in the hypertensive heart [236]. The same authors also found miR-29b restoration to be therapeutic, as it prevents further fibrosis.

##### miR-22

miR-22 has been studied for its role in opposing Ang II-induced cardiac fibrosis, and it has been shown that its overexpression and inhibition resulted in reduced or enhanced collagen deposition in cultured cardiac fibroblasts [237]. Its direct target has been shown to be TGFBR1, pointing to the possibility that miR-22 can attenuate Ang II-induced TGF-β signalling and thereby reduce cardiac fibrosis [237].

##### miR-let-7i

miR-let-7i attenuates Ang II-induced cardiac inflammation and fibrosis through targeting IL-6 and collagens [238]. Its level was found to be reduced in mouse hearts 3 and 7 days after Ang II infusion [238]. The knockdown of miR-7i exacerbates Ang II-induced cardiac inflammation and fibrosis, while overexpression or delivery of miR-let-7i suppressed it [238].

### 3.3. miRNAs That Directly Target RAS Components

#### 3.3.1. miR-181a

miR-181a has been identified to target the renin gene *REN* and the apoptosis-inducing factor 1 (*AIFM1*) mRNA directly, along with miR-663 which also interacts with *REN* [239]. Recent studies point to the potential role of miR-181a in mediating the crosstalk between the sympathetic nervous system and RAS in hypertension regulation. In genetically hypertensive BPH/2J mice, overactivation of the sympathetic system also induces increased renin release through sympathetic hyperstimulation of the kidney, while miR-181a downregulation in this context is associated with increased renin synthesis [240]. miR-181a mimic injection reduced renal renin mRNA levels as well as inflammatory marker toll-like receptor 4 (TLR4) mRNA levels and reduced blood pressure in BPH/2J mice, but not in normotensive BPH/3J mice [241]. The mimic injection also abolished the blood pressure-reducing effect of enalaprilat, suggesting that the antihypertensive effect of miR-181a is mediated by RAS inhibition [241]. The level of miR-181a is likely to be under the inhibitory influence of sympathetic stimulation, as renal denervation increased miR-181a levels [241]. miR-181a was also found to be one of the TLR4-responsive miRNAs which is downregulated in patients with coronary artery disease (CAD), along with miR-31, miR-16 and miR-145, making it a promising biomarker candidate [242]. A 12-month treatment with a combination of ARB (telmisartan) or ACE inhibitor (enalapril) with atorvastatin elevated those four miRNAs, with ARB having a more significant effect [242].

In summary, miR-181a directly reduces renin mRNA levels to attenuate RAS tone. The sympathetic system activation can inhibit miR-181a expression and thus increase RAS activity, contributing to blood pressure elevation. miR-181a could also serve as a biomarker for the monitoring of CAD and its pharmacotherapy.

#### 3.3.2. miR-143/145 Cluster

The miR-143/145 cluster is responsible for VSMC phenotypic switching, which refers to the dedifferentiation of contractile VSMC cells to adapt to a proliferative/biosynthetic state. VSMC phenotypic switching is important for atherosclerosis, intimal hyperplasia and hypertension in multipotent cardiac progenitor cells and later in the VSMCs [243]. miR-143 and miR-145 are directly targeted transcriptionally by SRF, myocardin and NKX2.5, which are important for VSMC phenotype and cardiac morphogenesis [243,244]. The two miRNAs cooperatively target other transcriptional factors important for VSMC differentiated states, including Kruppel-like factor 4/5 (KLF4/5), myocardin and ELK-1 [243,245,246]. Importantly, the VSMCs of miR-143/145-knockout mice were unable to switch their phenotype from a proliferative/biosynthetic state to a contractile state [246]. In these miR-143/145-deficient mice, ARBs and ACE inhibitors partially reversed vascular dysfunction, indicating the participation of RAS signalling in pathogenesis [243]. miR-145, along with miR-27a and miR-27b, also targets ACE [246,247]. In cultured VSMCs, mechanical stretch activates ERK1/2 and ACE expression, leading to the attenuation of miR-145 levels, presumably through elevated Ang II levels [248]. On the other hand, miR-143 levels are reduced in relatively de-differentiated proliferating atherosclerotic vessels, and its plasma level is negatively correlated to blood pressure, showing a significant reduction in essential hypertension [243,249]. Furthermore, miR-143 and miR-421 are involved in the modulation of the cardioprotective ACE2/Ang (1-7)/MasR axis of RAS via targeting ACE2 [34,250,251].

#### 3.3.3. miR-483-3p

A luciferase assay has revealed that miR-483-3p targets angiotensinogen, ACE, ACE2 and AT2R mRNAs [252]. Especially, in VSMCs, the expression levels of angiotensinogen and ACE induced by AT1R are specifically modulated by miR-483-3p [252].

#### 3.3.4. miR-766

The Cyp11B2 gene codes for aldosterone synthase, the rate-limiting enzyme for aldosterone synthesis, and it is a target of Ang II. The human Cyp11B2 gene possesses 735A > G polymorphism and −344T > C polymorphism that are in linkage disequilibrium, and the −344T allele is associated with hypertension [253]. In individuals with the −344T allele, miR-766 binds to the 735G-allele of Cyp11B2 gene and reduces aldosterone synthase mRNA levels in the Ang II-responsive human adrenocortical cell line H295R [253].

#### 3.3.5. miR-125b

In renal tubular epithelial HK-2 cell lines, miR-125b was shown to inhibit ACE2, mediating ROS production and apoptosis induced by high glucose, as the overexpression and knockdown of miR-125b lead to elevated and depressed ACE2 levels, respectively [254].

**Table 1 ijms-22-04762-t001:** Summary of miRNAs in RAS-induced cardiac remodelling.

miRNA	Target	Function	Source
miR-155	Socs1, Jarid2, AT1R, IKKi eNOS	Promote pro-hypertrophic and pro-fibrotic inflammation, but anti-hypertrophic through targeting AT1R	[90,91,92,95,97]
miR-208	Thyroid hormone-associated protein 1, myostatin, NLK	Promote cardiac hypertrophy and arrhythmia, involved in apoptosis	[104,109]
miR-132/212	FoxO3, AT1R, AC, JAK2, PKC, cJUN, SOD2, EGR1	Promote cardiac hypertrophy, inhibit autophagy and apoptosis	[112,114]
miR-21	Spry1, PTEN, SMAD7, TGFBR2, RECK	Promote cardiac hypertrophy and fibrosis	[116,201,202,203,204,206,210,212]
miR-410/495	MDM2, MET, MTA3, MTA1A, SOX9, BMI1,TBC1D9, PPX3, MEIS1, ATP7, PTP4A3, SMR3B	Promote cardiomyocyte hypertrophy and proliferation	[118]
miR-19a/b	Atrogin-1, MuRF-1, PTEN	Promote cardiac hypertrophy, inhibit apoptosis, reduce CTGF expression	[119,120,121]
miR-20b, 20b-5p	Mitofusin2, SMAD7	Promote cardiac hypertrophy and ventricular remodelling	[123,125]
miR-21-3p	SORBS2, PDLIM5, HDAC8	Involved in cardiac hypertrophy (evidence for both pro- and anti-hypertrophic effects)	[126,128]
miR-26a	GATA4, CTGF, collagen I	Inhibit cardiac hypertrophy and cardiac fibrosis	[127,228]
miR-16	Cyclin D1, D2, E1	Inhibit cardiac hypertrophy	[129]
miR-98	Cyclin D2	Inhibits cardiac hypertrophy	[130]
miR-30a	Beclin-1	Inhibits cardiac autophagy and hypertrophy	[131]
miR-34a	ATG9A, SMAD4	Inhibit cardiac autophagy and hypertrophy, promote cardiac fibrosis	[132,224]
miR-133a	Angiotensinogen, SRF, cyclin D2, caspase 9, CTGF, RhoA, Cdc42, Nel-A/WHSC2, collagen 1α1, NFATc4	Inhibit cardiac hypertrophy and cardiac fibrosis	[135,141,142,143,144,145,146,147]
miR-1	HDAC4, NFATc3, MCU, CDK6, IGF-1, IGF-1R	Inhibit cardiac hypertrophy	[141,152,153]
miR-99a	mTOR1/2	Inhibits cardiac hypertrophy	[157]
miR-101	Rab1a	Inhibits cardiac hypertrophy	[158]
miR-129-3p	PKIA	Inhibits cardiac hypertrophy	[159]
miR-433	JNK1, AZIN1	Promote cardiac fibrosis	[215]
miR-503	Apelin-13	Promotes cardiac fibrosis	[216]
miR-19a/b-3p	TGFBR2, SMAD4	Inhibit cardiac fibrosis	[230,231,232]
miR-29b	Collagen, fibrillin, elastin	Inhibit cardiac fibrosis	[233,234,235]
miR-22	TGFBR1	Inhibits cardiac fibrosis	[237]
miR-let-7i	IL-6, collagens	Inhibit cardiac inflammation and fibrosis	[238]
miR-181a	REN, AIFM1	Reduce blood pressure	[239]
miR-143/145	KLF4/5, myocardin, ELK-1	Cardiogenesis, maintenance of VSMC phenotype	[243,245,246]
miR-145, 27a/b	ACE	RAS modulation	[246,247]
miR-143, 421	ACE2	RAS modulation	[34,250,251]
miR-483-3p	ACE, ACE2, AT2R	RAS modulation	[252]
miR-766	Cyp11B2	Reduces aldosterone synthase	[253]
miR-125b	ACE2	RAS modulation	[254]

## 4. Conclusions

The RAS plays a crucial role in adaptive and detrimental cardiac remodelling. The effects of RAS on cardiac remodelling emerge as a balance between two opposing arms of RAS. On one arm, mediated primarily by PRR and AT1R, the cellular response to the cascade favours cardiac inflammation, hypertrophy and fibrosis, resulting in compromised cardiac function in the long run. The other cardioprotective arm primarily consists of alternative cleavage products of Ang I and Ang II, mediated through different sets of GPCRs, namely AT2R, AT4R, MasR, MrgD, the Ac-SDKP receptor and APJ. However, the mutual cooperation of both systems seems essential to control and calm excessive remodelling, oxidative stress and exaggerated inflammation.

miRNAs have emerged as novel regulators of RAS-induced cardiac remodelling by targeting and silencing the mRNA of RAS compounds including receptors and signalling molecules. miRNA expression is altered in response to cardiac injury or Ang II stimulation, and the modulation of their levels was potentially able to promote or suppress cardiac remodelling, suggesting their potential as therapeutic targets.

In cardiac hypertrophy, Ang II exerts its effect primarily on cardiomyocytes through AT1R. miRNAs are involved in the modulation of all three major components of AT1R signalling in cardiac hypertrophy, namely the canonical pathway, the EGFR transactivation pathway, and the inflammatory pathway. Pro-hypertrophic miRNAs, specifically miR-155, miR-208, miR-132, miR-212, miR-21, miR-410, miR-495, miR-19a/b, and miR-20b augment pro-hypertrophic signalling through silencing anti-hypertrophic proteins or targeting inhibitors of ERK/AKT pathways or inducing inflammation. Anti-hypertrophic miRNAs, specifically miR-21-3p, miR-26a, miR-16, miR-98, miR-30a, miR-34a, miR-133a, miR-1, miR-99a, and miR-129-3p exert their effects through the inhibition of various Ang II-induced pro-hypertrophic signalling molecules, transcription factors, or autophagy.

Ang II applies its signalling function not only in cardiomyocytes but also in cardiac fibroblasts and infiltrating immune cells. Ang II-induced ROS from NOX2, NOX4 or mitochondrial NLRP3 activates a myriad of signalling pathways that transform fibroblasts into myofibroblasts and favour ECM deposition. TGF-β signalling is also indispensable for the pro-fibrotic action of Ang II, in both canonical pathways through the action of SMAD proteins as well as non-canonical pathways including TAK1 activation. Pro-fibrotic miR-21, miR-433, miR-503, miR-34a and miR-155 increase fibrosis-mediating ERK, p38, AKT and SMAD pathways signalling or attenuating the cardioprotective action of Apelin. Contrarily, anti-fibrotic miR-26a, miR-133a, miR-19a/b-3p, miR-29b, miR-22 and miR-let-7i can attenuate TGF-β signalling, reduce cardiac inflammation or directly target ECM protein mRNAs.

Finally, RAS tone can also be under the direct modulation of miRNAs. Renin is targeted by miR-181a, while ACE is targeted by miR-145 and miR-27a/b, while ACE2 is targeted by miR-143, miR-421 and miR-125b. miR-483-3p targets angiotensinogen, AT2R, as well as both ACE and ACE2. The aldosterone synthase enzyme with 735G polymorphism is targeted by miR-766. Thus, it is not unreasonable to suggest that RAS-miRNAs interplay could be a bidirectional process with a mutually positive or negative feedback mechanisms.

Our understanding of miRNA in RAS is still expanding, and the miRNAs listed above by no means represent a comprehensive picture of all miRNAs involved in RAS-induced cardiac remodelling. Future investigation will hopefully disclose the function of the biologically most important miRNA-players in cardiac hypertrophy, necrosis/apoptosis, inflammation and fibrosis. Information about the aberrant expression of miRNAs in particular cardiac injuries or pathological alterations could serve as novel biomarkers for diagnostics, and the enhanced expression or inhibition of miRNAs could hold great therapeutic promise in preventing or reversing cardiac remodelling.

## Figures and Tables

**Figure 1 ijms-22-04762-f001:**
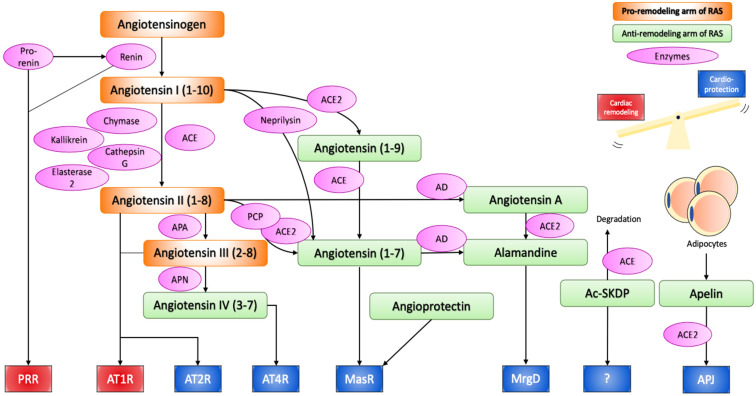
Renin–angiotensin system.

**Figure 2 ijms-22-04762-f002:**
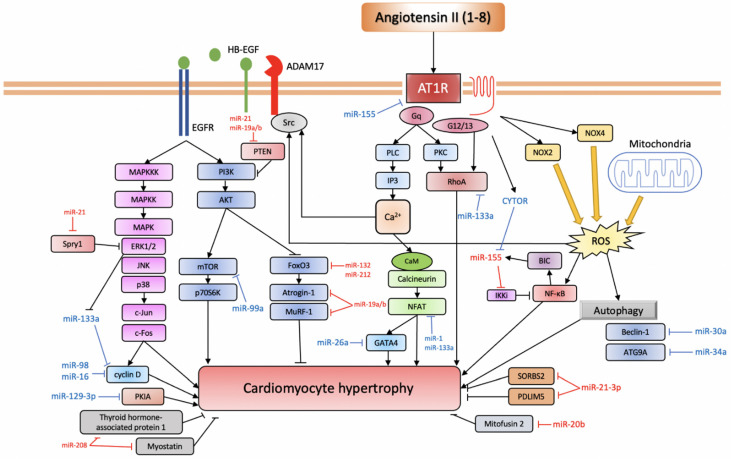
miRNAs involved in Ang II-induced cardiac hypertrophy.

**Figure 3 ijms-22-04762-f003:**
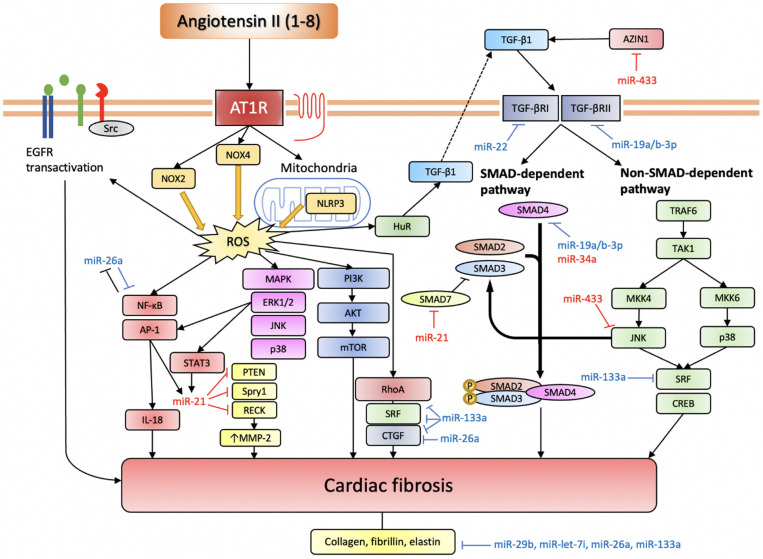
miRNAs involved in Ang II-induced cardiac fibrosis.

## Data Availability

Not applicable.

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
