# Peer review of "The Impact of microRNAs in Renin–Angiotensin-System-Induced Cardiac Remodelling"

_ijms, 2021, doi:10.3390/ijms22094762_

Round 1

Reviewer 1 Report

In their review "The impact of microRNAs in renin-angiotensin system-induced cardiac remodelling", Adamcova et al. present a comprehensive overview on the renin angiotensin system and its potential harmful effects in cardiomyocytes with special focus on miRNA-associated effects. The review is well structures and comprises several figures which give a good overview. Although miRNA biology is not totally novel, there is no such overview yet and this concise overview totally qualifies for publication in IJMS.

Author Response

We are thankful  for very positive review of our manuscript. The revision of English has been done by the native speaker, expert in medicine. In addition, we have included 3 following citations recommended by Editor Office and all citations have been renumbered.

  1. Deiuliis, J.; Mihai, G.; Zhang, J.; Taslim, C.; Varghese, J.J.; Maiseyeu, A.; Huang, K.; Rajagopalan, S. Renin-sensitive microRNAs correlate with atherosclerosis plaque progression. J Hum Hypertens 2014, 28, 251-258, DOI:10.1038/jhh.2013.97.
  2. Butterworth, M.B. Role of microRNAs in aldosterone signaling. Curr Opin Nephrol Hypertens 2018, 27, 390-394, DOI:10.1097/MNH.0000000000000440.
  3. Butterworth, M.B. Non-coding RNAs and the mineralocorticoid receptor in the kidney. Mol Cell Endocrinol 2021, 521, 111115, DOI:10.1016/j.mce.2020.111115.

Enclosed please the revised manuscript of the review titled “The impact of microRNAs in renin-angiotensin system-induced cardiac remodelling”.

Reviewer 2 Report

The review article by Adamcova et al gave a substantial introduction and detailed review regarding the impact of microRNAs in RASS-induced cardiac remodeling. This is a well-written review and will provide insights for scientists who focus on this area.

Author Response

(The authors gave the same response as above.)
